# Evanescent field trapping and propulsion of Janus particles along optical nanofibers

Georgiy Tkachenko [1] ✉, Viet Giang Truong [1], Cindy Liza Esporlas[1], Isha Sanskriti [1] & Síle Nic Chormaic [1] ✉

Small composite objects, known as Janus particles, drive sustained scientific interest primarily targeted at biomedical applications, where such objects act as micro- or nanoscale actuators, carriers, or imaging agents. A major practical challenge is to develop effective methods for the manipulation of Janus particles. The available long-range methods mostly rely on chemical reactions or thermal gradients, therefore having limited precision and strong dependency on the content and properties of the carrier fluid. To tackle these limitations, we propose the manipulation of Janus particles (here, silica microspheres half-coated with gold) by optical forces in the evanescent field of an optical nanofiber. We find that Janus particles exhibit strong transverse localization on the nanofiber and much faster propulsion compared to all-dielectric particles of the same size. These results establish the effectiveness of near-field geometries for optical manipulation of composite particles, where new waveguide-based or plasmonic solutions could be envisaged.

The term "Janus particle" (JP), inspired by the ancient Roman two-faced god, refers to a composite micro- or nanoscale artificial object having multiple parts with distinct chemical or physical properties. Introduced in 1988 by C. Casagrande et al.[1] and popularized by P.-G. de Gennes in his 1991 Nobel Prize lecture on Soft Matter[2], the JP concept sparked intense research activity which led to numerous applications, primarily in the biomedical domain because of the versatility and size-wise compatibility of such particles with living tissue[3,4]. Some notable examples include the use of JPs for unidirectional association with human endothelial cells[5], magnetolytic therapy for cancer[6], specific cellular targeting, sensing, and spectroscopy[7], and biomarking for detection via dark-field imaging[8]. Although the most common geometry for a JP is a sphere with distinct halves, many other kinds are possible, such as dimers[9], nanocorals[7], nanotrees[10], and even homogeneous particles having transient "Janus" properties induced by an external field[11,12]. Typically, a JP is produced via nanofabrication, by incorporating a nanoscale metallic coating or inclusion in a homogeneous carrier particle, usually a commercially available dielectric bead.

Naturally, one would wish to have a precise tool for handling JPs, similar to what optical tweezers provide for contactless manipulation of various small-scale objects[13]. After early tests in optical tweezers[14], it was soon realised that JPs with metallic parts are very hard to handle by light, because the high reflectance and absorbance of the metal lead to strong optical and thermal forces repelling the particle from the optical trap. For instance, JPs in water were seen to escape from higher-intensity regions and settle around higher-gradient ones[15,16]. It was also noted that—precisely because of their inhomogeneity—JPs commonly exhibit various forms of "taxis", that is, motion in response to external stimuli, such as temperature[15–20], diffusion[21,22], magnetic fields[23], or chemical reactions[9–11,23–26]. Phototaxis via thermophoretic forces emerged as one of the most promising methods, because it enables the particles to be moved by light, without the need for any chemical fuel. However, these forces are very weak in common environments (e. g., water) and fail to produce a considerable speed. To enhance thermophoresis, and thus render a JP an efficient microswimmer or micro-machine, major attention has been devoted to studies on metallo-dielectric JPs in near-critical mixtures of liquids where the system's behavior is at its most sensitive to minute changes in temperature or pressure[12,19,21,27,28].

Unrestricted JPs assume random orientations in the host environment. Therefore, special arrangements are needed in order to make

[1]Light-Matter Interactions for Quantum Technologies Unit, Okinawa Institute of Science and Technology Graduate University, 1919-1 Tancha, Onna-son 904-0495 Okinawa, Japan. ✉e-mail: georgiy.tkachenko@oist.jp; sile.nicchormaic@oist.jp

the induced motion directional. In particular, one can use confined environments[21,22], "polarization" in thermal gradients[18], gravity[29], structural defects[25,30], magnetic fields[23], elastic forces[31], multiple programmable light sources[10,11] or structured fields[19]. With optical techniques being the most promising, until now there have been no efficient methods for fast and precise manipulation of JPs via optical forces alone. Such a method would be very beneficial for many applications of JPs, because purely optical manipulation does not rely on chemical processes or special conditions such as temperature distributions, exotic mixtures, or fluidic backgrounds.

Recalling that metallo-dielectric JPs in optical fields tend to move toward higher-gradient areas[15,16], a logical strategy is to use evanescent fields, where the intensity exhibits a steep gradient near the surface, thereby constraining the particle in the radial direction while allowing it to move along the wave vector. There have been numerous works on optical manipulation via evanescent fields in the vicinity of a glass prism[32], planar waveguide[33], and more recently, an optical nanofiber (ONF), which proved to be very effective in producing propulsion[34–38], binding[36,39–41], rotation[42], detection[43], and sorting[44,45] of various kinds of micro- and nanoparticles, but never metallo-dielectric ones.

Here, we report on nanofiber-mediated all-optical trapping and propulsion of JPs, in this case silica microspheres half-coated with a few-nanometer-thick layer of gold (Fig. 1). In contrast to earlier studies with free-space beams of light, evanescent fields allow one to achieve stable trapping of JPs, without a high limit for the optical power. Furthermore, the presence of the metallic coating results in significant enhancement of the particle's propulsion speed, compared to an all-dielectric particle of the same size. Near-field manipulation truly benefits from both "faces" of the JP, featuring the trapping ability and thermal neutrality of a dielectric object, and the high polarizability of a metal.

## Results

### Optomechanical model

We consider a silica-gold JP in water near an ONF that guides a fundamental, plane-polarized mode. Since both the JP and the ONF+mode are mirror-symmetric objects, we expect the combined light-matter system in its stationary state to be symmetric with respect to the polarization plane, given the gravitational acceleration, **g**, also lies in this plane. Therefore, a simplified 2D model might seem sufficient for simulation of this optomechanical system. However, such a model treats the gold coating as a half-cylinder instead of a half-dome cap,

thereby leading to a significant deviation from the real physical picture. Consequently, we have to use a full 3D numerical model, with its cross-section through the symmetry plane sketched in Fig. 2a. Here, the mode propagates along the fiber axis, $z$, the input laser beam has an electric field vector, $\mathcal{E}$, lying in the $yz$ plane, and **g** is parallel to $y$. The system has the following parameters: the mode has the vacuum wavelength of $\lambda = 1.064\,\mu m$ and the power of $P = 1\,mW$; the ONF and the particle have radii of $R_f = 0.35\,\mu m$ and $R_p = 1.5\,\mu m$, respectively, and are surrounded by water at normal ambient conditions; one hemisphere of the particle is coated with a thin uniform layer of gold over a 5-nm-thick titanium adhesion layer[46].

If only optical action is considered, the particle is pulled to the ONF by the radial force, $\mathbf{F}_y$, and pushed along $z$ by the longitudinal force, $\mathbf{F}_z$. These forces are simulated by integration of the Maxwell's stress tensor over the surface enclosing the particle. Optical forces acting on the surface elements lead to the optical torque, $\mathbf{N} = \mathbf{N}_x$, and the particle spinning in the $yz$ plane. As shown in Fig. 2b, $\mathbf{N}_x < 0$ for any orientation of the gold cap, $\alpha$, therefore the spinning is expected to be continuous. However, $\mathbf{N}_x$ is opposed by the mechanical torque due to the dynamic friction force, $\mathbf{F}_{fr}$, with a magnitude of $-\mu F_y$, where $\mu$ is the friction coefficient. To find this coefficient theoretically, one needs to solve the opto-hydrodynamic problem while taking into account that the particle-fiber contact is intermittently broken by thermal motion and the inertial lift in a shear flow[47]. Since this study focuses on light-induced dynamics of JPs, we find $\mu$ by fitting the theoretical balance orientation, $\alpha_0$, to its experimentally measured values (see Fig. 3c). This orientation occurs when the total torque,

$$N_\Sigma = N_x + N_{fr} = N_x - \mu F_y R_p \tag{1}$$

crosses zero with a negative slope (see Supplementary Note 1). Interestingly, $\alpha_0$ is nearly independent of the gold cap thickness, $d$, which otherwise strongly influences the optomechanics of JPs.

In order to straightforwardly connect the forces and the measurable dynamics of JPs and silica particles (SP), we define dimensionless enhancement factors,

$$\xi_{i=yz} = F_{JPi}/F_{SPi} \approx V_{JPi}/V_{SPi} \tag{2}$$

where $V_{JP,i}$ and $V_{SP,i}$ are the speeds of the particles' center-of-mass. The approximate equality in Eq. (2) relies on the assumption that SPs and

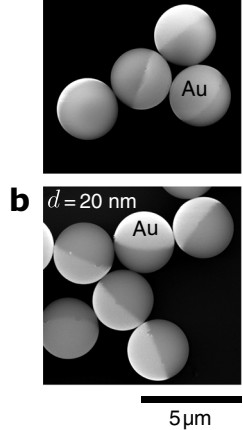

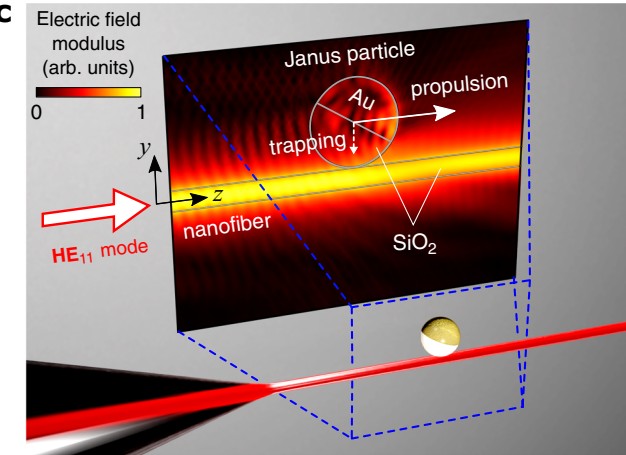

**Fig. 1 | Janus particles and the nanofiber-based manipulation concept. a, b** Scanning electron microscope (SEM) images of silica-gold JPs used in this study ($d = 10$ nm and 20 nm is the gold coating thickness). **c** Artistic view of a JP in contact with the waist region of an ONF guiding a fundamental mode, **HE₁₁**, polarized in the symmetry plane of the ONF-JP system ($yz$-plane). The electric field distribution

(same as in Fig. 2e) is simulated for the wavelength of 1.064 μm, 0.7-μm-thick ONF, 3-μm particle, 20-nm-thick gold cap tilted forward at an angle $\alpha = 30°$ with respect to $y$. The particle is propelled along $z$ by the scattering optical force, while staying trapped in the $y$ direction due to the attractive gradient force.

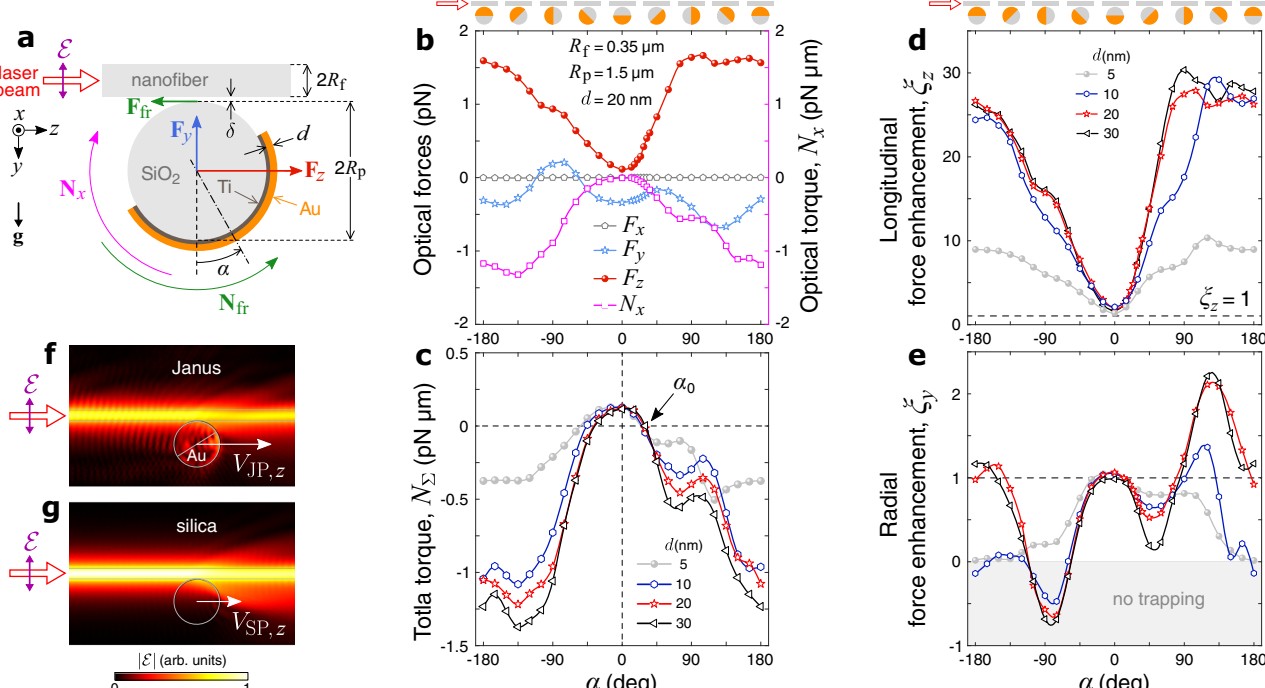

**Fig. 2 | Numerical simulations of optical fields, forces, and torques in the particle-nanofiber system. a** Model schematics, where $\mathcal{E}$ is the electric field vector; **g** is the gravitational acceleration; $\mathbf{N}_x$ is the optical torque; $\mathbf{N}_{\mathrm{fr}}$ is the torque produced by the friction force, $\mathbf{F}_{\mathrm{fr}}$; $\mathbf{F}_z$ and $\mathbf{F}_y$ are the longitudinal and radial optical forces, respectively. **b** Optical forces and torque versus the gold cap orientation for different coating thickness, $d$. **c** Total torque, $N_\Sigma = N_x + N_{\mathrm{fr}} = N_x - \mu F_y R_{\mathrm{P}}$. **d, e** Force enhancement factors, $\xi_z = F_{\mathrm{JP},z}/F_{\mathrm{SP},z}$ and $\xi_y = F_{\mathrm{JP},y}/F_{\mathrm{SP},y}$. Markers in (**b–e**) indicate the simulated points, lines refer to spline interpolations. Sketches on top show the gold cap orientation at every 45 degrees. Electric field distributions around (**f**) Janus ($d = 20$ nm, $\alpha = 30°$) and (**g**) silica particles indicate stronger back-scattering of light in the JP case (compare the bottom-left parts of the field maps), which explains its higher propulsion speed, $V_{\mathrm{JP},z} > V_{\mathrm{SP},z}$.

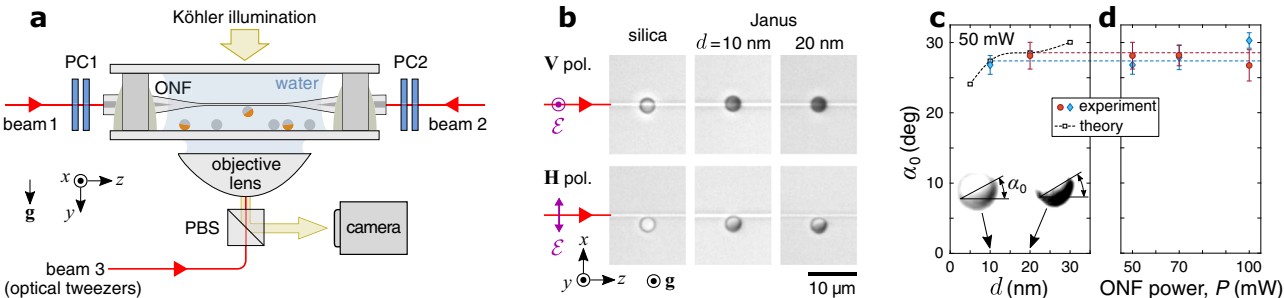

**Fig. 3 | Experimental details. a** Schematic of the optical setup (not to scale). A single-mode ONF is immersed in a water suspension of 3 μm silica and Janus particles which are selectively trapped and delivered to the ONF waist by low-power (1.3–1.7 mW) optical tweezers. Laser beams 1 and 2 produce counterpropagating modes polarized in $xz$- or $yz$-planes, corresponding to **H** or **V** polarization states of the beams. To ensure the polarization is maintained upon propagation to the ONF waist, we use adjustable compensators, PC1 and PC2, consisting of two quarter-wave plates each. The ONF and the particles are imaged by a CMOS video camera through the trapping objective under white-light illumination in Köhler

configuration; PBS is a polarizing beamsplitter. **b** Particles are seen trapped in the polarization plane. JPs with 10- and 20-nm-thick coating were studied in two different samples, both ONFs having $R_{\mathrm{f}} = 0.35$ μm as measured by SEM; the transmitted optical power through the fiber was 50 mW. **c, d** Measured (markers with error bars respectively indicating the mean values and 50% standard deviation ranges obtained for 3-5 different JPs) and simulated (markers connected via a spline interpolation) stable orientations of JPs vs. the gold thickness and the optical power transmitted through the ONF. Insets: particle images with enhanced lightness and contrast.

JPs have nearly identical geometries and that the speed is proportional to the driving force (in the limit of small Reynolds number as is the case in this work). The latter statement implies that thermal changes to the fluid's viscosity are negligible.

For each given thickness the propulsion enhancement, $\xi_z$, is maximal when the gold cap is oriented toward the fiber ($|\alpha| \to 180°$). This result follows from the high polarizability of gold and the sharp decline in the evanescent field with distance from the fiber surface. In the explored range of gold thicknesses from 5 to 60 nm, the maximum $\xi_z$ approaches 30 and saturates around $d = 30$ nm, which is close to the skin depth of gold[48]. When the cap is facing away from

the fiber ($|\alpha| \approx 0°$), the enhancement is close to unity, meaning that the gold layer in this case has only a minor influence on the light-induced propulsion of the particle. When a JP assumes its stable orientation, $\alpha_0 \approx 30°$ for $\mu = 0.25$, the propulsion is expected to be enhanced by a factor of $\xi_z \lesssim 10$. To illustrate the origin of this enhancement, we plot the simulated electric field distributions (Fig. 2f, g) where more intense back-scattering (hence, larger transferred linear momentum toward $+z$) is seen for a JP compared with a SP. Given this effect, an ONF-propelled JP calls for an analogy with a vessel pushed by the pressure of wind on its sail; gold-coated (with "sail") silica particles tend to move very similarly to uncoated

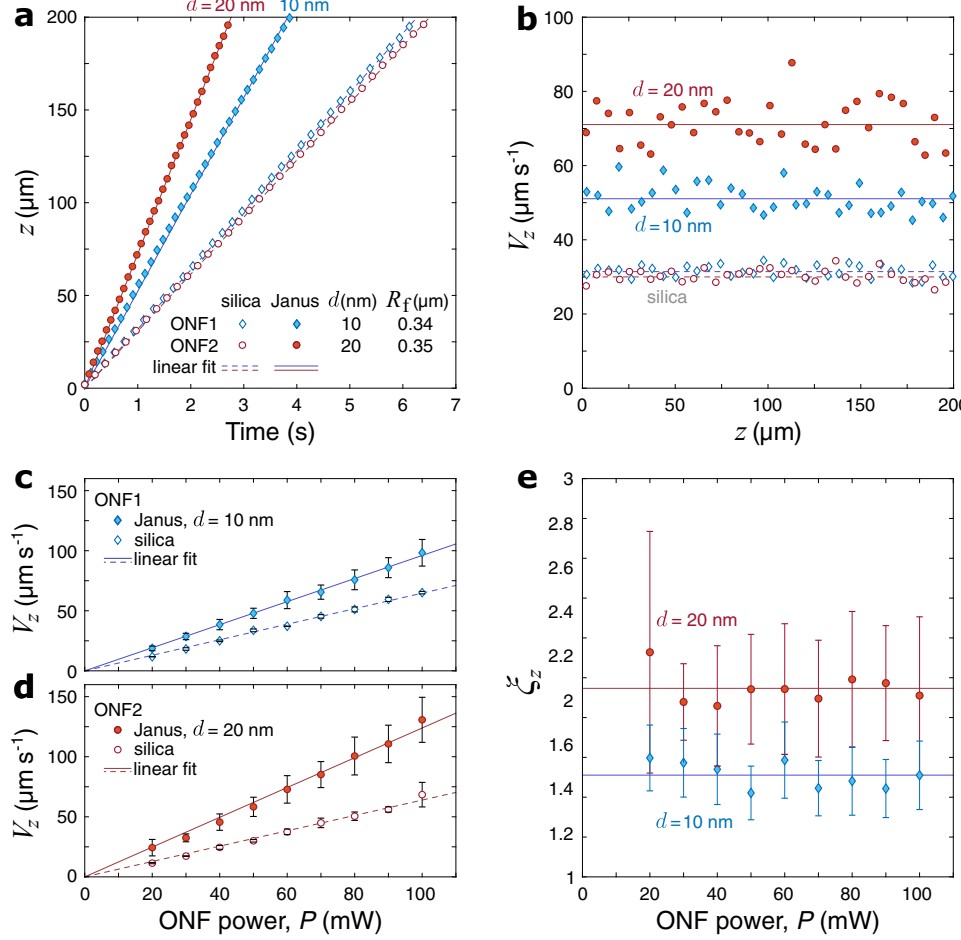

**Fig. 4 | Measured optomechanics of silica and Janus particles near an ONF.** Input beam 1 was **V**-polarized, and we used dedicated samples labeled as ONF1 and ONF2 for the gold coating thickness of 10 nm and 20 nm, respectively (see Supplementary Movie 2). **a** Sample position-time series for the longitudinal direction of the particle motion at a transmitted optical power of 50 mW. Markers indicate the $z$ coordinate of the particle's center (with respect to that in the first frame); lines refer to the best linear fit to the data. **b** Propulsion speed as a function of the particle's position along the ONF waist. Lines indicate the mean speed values which are (in units of $\mu m\ s^{-1}$): $\langle V_{JP,z}\rangle|_{ONF1} = 51.1$, $\langle V_{SP,z}\rangle|_{ONF1} = 31.4$, $\langle V_{JP,z}\rangle|_{ONF2} = 71.1$, and $\langle V_{SP,z}\rangle|_{ONF2} = 30.0$. **c, d** Propulsion speed, $V_z$, is proportional to the transmitted optical power. **e** Propulsion enhancement factors, $\xi_z$, are nearly constant throughout the explored range of power, averaging at (horizontal lines) $1.51 \pm 0.16$ and $1.95 \pm 0.34$ for 10-nm- and 20-nm-thick coating, respectively. Markers and error bars in (**c**–**e**) respectively indicate the mean values and the 50% standard deviation ranges obtained for a sample of 5–7 different JPs or 2–6 different SPs for each power value measured in transmission through the ONF.

(no "sail") particles, except the "sail" gives a significant gain in the longitudinal speed.

For the radial direction, the maximum enhancement is much smaller than for the longitudinal one. Some orientations of the cap give $\xi_y \ll 1$ or even $\xi_y < 0$ (for larger $d$ values and $\alpha \approx -80 \pm 30°$), in which case the radial optical force on the particle is repelling it from the fiber.

To test the above predictions experimentally, we chose two representative values of gold thickness, 20 and 10 nm. The former is expected to produce the strongest propulsion enhancement, while the latter provides a quantitative verification of the model by allowing us to compare two distinct cases. The choice of $d = 5$ nm would provide even higher distinction; however, in practice such thin gold coatings appeared patchy following JP fabrication and therefore were not suitable for making comparisons with the model.

**Experiment**

Since the optomechanical behavior of the JPs was considered in relation to that of SPs, we performed each set of measurements using a mixture of both coated and uncoated particles interacting with the same short (~ 0.1 mm) stretch of the waist region of an ONF (see Fig. 3a and Supplementary Figure 6). The original plan was to disperse particles in heavy water ($D_2O$) since it has negligible absorption at the

working wavelength. However, this choice proved impractical, because JPs in $D_2O$ kept escaping from our single-beam optical tweezers soon after being lifted by about 20 μm from the bottom glass slide. In contrast, in Milli-Q water, we found that a JP was quickly attracted to the beam focus and hovered around it under the combined action of optical, gravitational, and thermophoretic forces, of which the last evidently required the fluid (not only the particle) to have some absorption. Even in this case, JPs could be stably trapped and delivered to the ONF only when the optical power in the tweezers did not exceed 2 mW (see Supplementary Movie 1); otherwise the particle wandered further from the focus and eventually escaped. It was particularly challenging to manipulate JPs in the $xz$-plane, because of the weakness of the optical gradient forces. Aside from the darker region of the gold cap (Fig. 3b, c), we used this unusual behavior as a clear indication that a given particle was a JP rather than a SP, which predictably exhibited stable optical trapping with the trap stiffness being proportional to the optical power.

When a JP is propelled under horizontal polarization (lower row in Fig. 3b), the gold cap exhibits a noticeable tilt in the direction of propagation, as verified by analyses of the captured images (see Supplementary Fig. 7). The tilt remains constant while the particle's motion is uniform, which confirms the theoretically predicted stable orientation

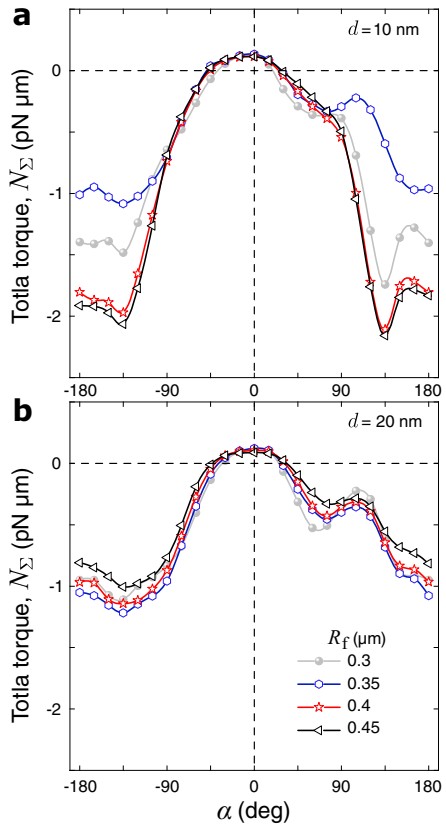

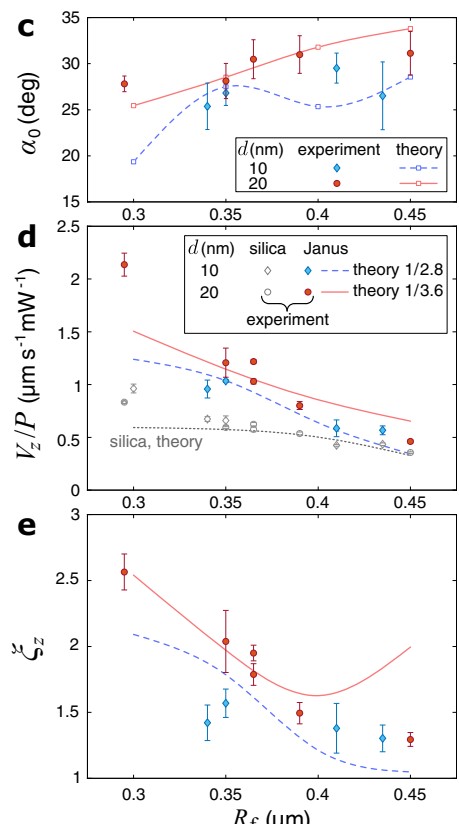

**Fig. 5 | Validation of theoretical model. a, b** Simulated torque on a JP versus the gold cap orientation for various ONF radii, $R_f$. **c** Measured (markers with error bars) and simulated (markers connected via a spline interpolation) balance orientation as a function of $R_f$ for 10- and 20-nm-thick gold coating. **d** Measured (markers with error bars) and simulated (lines) propulsion speed (reduced by the transmitted ONF power, $P = 50$ mW) for Janus and silica particles, as a function of the ONF radius. **e** Corresponding propulsion enhancement factors. Simulated $V_{JP,z}$ were scaled down by the factors indicated in the legend. Markers and error bars in (**d**) and (**e**), respectively indicate the mean values and the 50% standard deviation ranges obtained for 5 different JPs or 2–5 different SPs recorded in each ONF sample with $R_f$ measured by SEM.

of a JP at an angle of $\alpha_0$ (see Fig. 2c). The best fit of the theoretical $\alpha_0$ (at $N_\Sigma = N_x - \mu F_y R_p = 0$) to the measured values (Fig. 3c) corresponds to a friction coefficient of $\mu = 0.25$. Since for a horizontally polarized mode the mirror symmetry of the light-matter system is broken by gravity ($\mathbf{g} \perp \mathcal{E}$), we used higher optical power ($P \geq 50$ mW) assuming that the gradient optical force is high enough to keep the particle in the $xz$-plane without significant sagging along $y$. Since both $N_x$ and $F_y$ scale with $P$, $\alpha_0$ is expected to be power-independent. This prediction agrees well with the experimental results, as shown in Fig. 3d. The error bars in Fig. 3c, d refer to the 50% standard deviation ranges obtained for a sample of 3-5 different JPs for each power value measured in transmission through the ONF.

Once trapped in the evanescent field, both silica and Janus particles exhibit steady propulsion along the ONF waist, with JPs consistently moving faster (see Fig. 4a and Supplementary Movie 2) at a nearly constant longitudinal speed (Fig. 4b). Therefore, our model assumption that thermal effects due to the gold cap are negligible is largely supported by the experimental evidence. Interestingly, as shown in Fig. 4c, d, the propulsion speed of JPs is proportional to $P$. Such behavior is typical for all-dielectric particles, but very unusual for metallo-dielectric JPs, which normally show dominant thermal motion due to strong heating and convection at higher intensities of light[16,29]. We conclude that the orientation of the gold cap pointing outward from the ONF (as confirmed by Fig. 3b, c and Supplementary Movie 3) results in the evanescent field interacting primarily with the dielectric core of the particle. At the same time, the presence of the cap results in a significant propulsion enhancement, that was nearly constant throughout the explored range of power, from 20 to 100 mW (Fig. 4e).

In the experiments presented herein, we limited the power to 100 mW because performing accurate particle manipulations at higher powers (hence higher speeds) was too challenging for our experimental protocol (see "Methods" for the details).

To account for the random size variation of the particles ($R_p \approx 1.5 \pm 0.1$ μm) and nanoscale imperfections of their coating (Fig. 1a, b), we characterized at least five different JPs and two SPs in each sample. Individual points in Fig. 4e are found as $\langle \xi_z \rangle = \langle V_{JPz} \rangle / \langle V_{SPz} \rangle$, for each power value. $\langle \ldots \rangle$ stands for averaging of the measured quantity (here, the mean propulsion speed) over the selection of particles. Error bars in Fig. 4e are calculated as $\pm \langle \xi_z \rangle \sqrt{[\sigma(V_{JPz})/\langle V_{JPz} \rangle]^2 + [\sigma(V_{SPz})/\langle V_{SPz} \rangle]^2}$, where $\sigma(\ldots)$ is the 50% standard deviation range.

## Analysis

Now let us consider how the measured particles' dynamics compare with the simulations. Since we did not observe any signature of thermal effects near the ONF, we assume that the key factor defining the optical force enhancement is the gold cap's balance orientation, $\alpha_0$, at which the total torque, $N_\Sigma$, crosses zero with a negative slope. Figure 5a, b shows $N_\Sigma(\alpha)$ simulated for nanofibers of various thickness values ($0.3 \leq R_f \leq 0.45$) which we targeted in the experiments. As one can see Fig. 5c, the theoretical values of $\alpha_0$ (found by spline interpolation of the simulated points near $N_\Sigma = 0$) are within the standard deviation ranges of most experimental points, each of which corresponds to image-based measurements on 3 to 5 different JPs (see Supplementary Fig. 7 for more details). However, the propulsion speed values (Fig. 5d) and enhancement factors (Fig. 5e) calculated

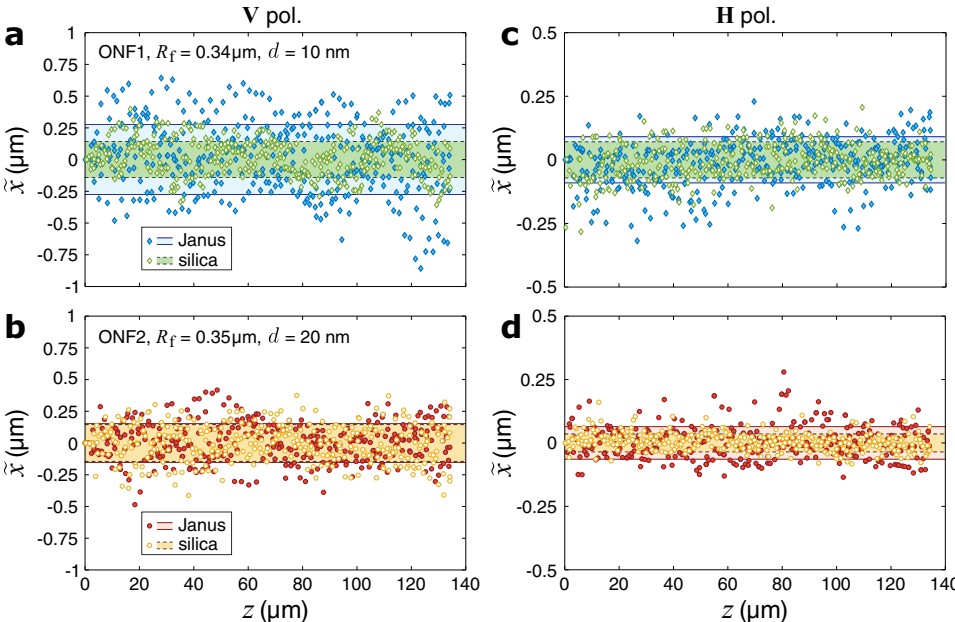

**Fig. 6 | Optical trapping dynamics of Janus and silica particles propelled by light near an ONF.** (Supplementary Movie 3) **a**, **b** Typical reduced trajectories of a JP compared to a SP in the same sample, with the ONF coupled to beam 1 with **V** polarization. **c**, **d** Same for **H**-polarized input. Optical power transmitted through the ONF was 70 mW, the gold thickness and ONF radius values are stated in the legends, the shaded bands mark the standard deviation ranges for the data.

for these $\alpha_0$ values turned out to be significantly higher than the experimental results for both 10- and 20-nm-thick coatings (by factors of approximately 2.8 and 3.6, respectively). This mismatch could be an artifact of the relatively coarse mesh we had to use in COMSOL to perform these 3D simulations at the limit of the available computational resources. Indeed, a similar mismatch (by a factor of 5–7) between the theory and the experiment was observed previously for simulations of non-spherical gold nanoparticles trapped in optical tweezers[49]. It is worth noting, the unscaled theoretical curve for the silica particles matches the data quite well (Fig. 5d). This fact verifies the accuracy of our model including the viscous drag estimation[42,47] which gives the propulsion speed as

$$V_z = F_z \left[ 6\pi\eta R_\mathrm{p} \left( \frac{8}{15} \ln \frac{\delta}{R_\mathrm{p}} - 0.9588 \right) \right]^{-1} \quad (3)$$

where $\eta$ is the dynamic viscosity of the fluid ($\eta = 1$ mPa s for water at room temperature) and $\delta = 10$ nm is the particle-fiber separation due to the surface roughness[50]. As one can see, the data point for JPs with $d = 10$ nm and $R_\mathrm{f} = 0.3$ μm is missing because in all our experiments with this geometry the JPs escaped from the fiber within ~10 μm of propulsion. We attribute this behavior to a greater impact of coating imperfections on the optomechanics of JPs when the gold layer is thinner and the evanescent field decays slower.

**Transverse localization**

As we have shown above, a JP stays trapped in the transverse direction while being stably propelled along the ONF. In order to study this 2D optical trapping quantitatively, we compared the random motion of silica and Janus particles along the $x$ axis. The recorded tracks of the particles are combined in Fig. 6, where $\tilde{x}(z) = x(z) - \langle x \rangle_z$ is the reduced transverse coordinate of the particle's center, and $\langle x \rangle_z$ is the mean transverse coordinate for each track. Following our assumption of thermal effects being negligible and assuming that the particle near an ONF is in a harmonic optical potential, we can estimate the strength of the transverse trapping based on the equipartition theorem[51]. It states that the trap stiffness (defined as the optical force per unit

displacement) can be found from $\kappa_x = k_B T / \sigma(\tilde{x})^2$, where $k_B$ is Boltzmann's constant, $T$ is the absolute temperature, and $\sigma(\tilde{x})^2$ is the particle's positional variance. Table 1 summarizes our analyses of the data sets shown in Fig. 6 by giving the measured propulsion enhancement factors, $\xi_z$, the standard deviation values for the transverse coordinate, $\sigma(\tilde{x}_{\mathrm{SP,JP}})$, the corresponding transverse enhancement factors, $\xi_x = [\sigma(\tilde{x}_{\mathrm{SP}})/\sigma(\tilde{x}_{\mathrm{JP}})]^2$, and the simulated radial enhancement factors, $\xi_r$ (equivalent to $\tilde{\xi}_y$ in Fig. 2e).

## Discussion

As shown in Table 1, both theory and experiments give $\xi_{r,x} < 1$ in all cases, that is gold-coated particles exhibit somewhat weaker transverse localization compared to uncoated ones. Strictly speaking, simulated $\xi_r$ can only be compared to $\xi_x$ measured under **H** polarization (bold values in Table 1), in which case the $x$ direction is truly radial for the 2D-trapped particle. Similarly to the case of longitudinal optical forces and the propulsion enhancement, our model overestimates radial optical forces and the transverse trapping strength. However, the difference between theoretical and experimental results for the radial direction is significantly smaller; compare $\xi_{z,\mathrm{theory}} \approx 3.58\,\xi_{z,\mathrm{experiment}}$ with $\xi_{r,\mathrm{theory}} \approx 1.33\,\xi_{x,\mathrm{experiment}}$ for ONF1 and $\xi_{z,\mathrm{theory}} \approx 3.98\,\xi_{z,\mathrm{experiment}}$ with $\xi_{r,\mathrm{theory}} \approx 2.53\,\xi_{x,\mathrm{experiment}}$ for ONF2. This asymmetry suggests that our model misses some non-optical

**Table 1 | Analysis of the data series shown in Fig. 6**

| Polarization | V | | H | |
|---|---|---|---|---|
| Sample | ONF1 | ONF2 | ONF1 | ONF2 |
| $R_\mathrm{f}$ (μm) | 0.34 | 0.35 | 0.34 | 0.35 |
| $d$ (nm) | 10 | 20 | 10 | 20 |
| $\xi_{z,\,\mathrm{experiment}}$ | 1.51 ± 0.12 | 1.88 ± 0.30 | 1.46 ± 0.13 | 1.78 ± 0.13 |
| $\xi_{z,\,\mathrm{theory}}$ | 5.23 | 7.09 | 5.23 | 7.09 |
| $\sigma(\tilde{x}_{\mathrm{SP}})$ (nm) | 142 | 144 | 71 | 35 |
| $\sigma(\tilde{x}_{\mathrm{JP}})$ (nm) | 277 | 152 | 91 | 64 |
| $\xi_{x,\,\mathrm{experiment}}$ | 0.26 | 0.90 | **0.61** | **0.30** |
| $\xi_{r,\,\mathrm{theory}}$ | 0.81 | 0.76 | **0.81** | **0.76** |

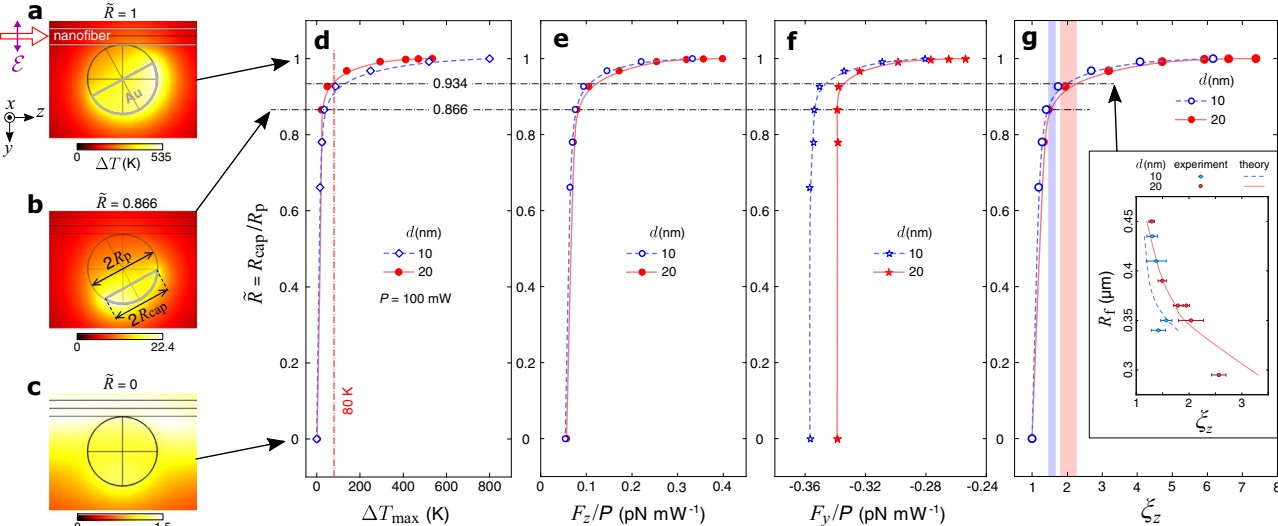

**Fig. 7 | Simulations with a reduced gold cap. a–c** Distributions of the temperature increase, $\Delta T = T - T_0$ (with respect to the room temperature of $T_0 = 20°C = 273.15$ K), around a Janus particle ($R_p = 1.5$ μm, $d = 20$ nm, $\alpha_0 = 28.5°$) near an ONF ($R_f = 0.35$ μm, $P = 100$ mW) in water. The gold cap has a variable base radius, $R_{cap}$; (**a**) same as in Fig. 2a ($R_{cap} = R_p$), (**b**) slightly reduced ($R_{cap} = 0.866 R_p$), (**c**) reduced to zero (no coating). **d** Maximum temperature increase (for $P = 100$ mW), (**e**) long-itudinal and (**f**) radial optical forces (reduced by $P$), (**g**) propulsion enhancement simulated for a Janus particle with a reduced gold cap, $\tilde{R} = R_{cap}/R_p \leq 1$, the two different coating thicknesses (10 and 20 nm with $\alpha_0 = 27.5°$ and $28.5°$, respectively), and a fixed ONF radius ($R_f = 0.35$ μm). Markers indicate the simulated points, solid and dashed lines refer to spline interpolations, shaded bands mark the standard deviation ranges for the experimental data. Inset in (**g**) shows the measured (markers with error bars, same data as in Fig. 5e) and the simulated (lines) pro-pulsion enhancement as a function of the ONF radius, $R_f$. The gold cap is simulated with $\tilde{R} = 0.934$ and the experimentally found values of the balance orientation angle, $\alpha_0$.

phenomena relevant for the radial direction. For instance, thermal forces are known to push silica-gold particles in water away from the higher-temperature region created around the cap[15,17]; thus Janus particles in our experiments might be pushed more strongly to the fiber surface than one would expect from optical forces alone.

Although thermophoresis in the JP-ONF system is beyond the scope of this study, we numerically simulated the temperature increase resulting from the conversion of electromagnetic energy into heat and its evolution in time (see Supplementary Note 2). We found that (i) the temperature distribution in the system becomes stable after less than 1 ms of evolution, therefore the particle's propulsion under optical forces can be safely neglected in thermal simulations; (ii) the temperature scales linearly with the optical power in the ONF; (iii) the predicted temperature increase, $\Delta T$, is unrealistically high, see Fig. 7a where the maximum of $\Delta T$ exceeds 500 K for a power of 100 mW. By contrast, cavitation bubbles have never been observed with a JP near an ONF in our experiments, which indicates that the boiling point of water has not been exceeded ($\Delta T \lesssim 80$ K). This mismatch suggests that our thermal model grossly overestimates the absorption of light by the gold cap. Indeed, the model considers the cap as a perfectly smooth film of crystalline gold, whereas in reality vapor deposition of gold onto microspheres is likely to produce polycrystalline coatings[52] with prominent nanoscale irregularities near the edges where the deposi-tion is increasingly oblique and may be affected by neighboring particles[30].

In order to mimic the decrease of absorption of light due to likely irregularities of the coating, we removed gold from areas beyond the effective cap radius of $R_{cap} \leq R_p$, see Fig. 7b. Since the evanescent field rapidly decays with distance from the ONF, the relative position of the cap's edge turns out to be crucial for the light-induced temperature increase (Fig. 7d) as well as the optical forces on the particle (Fig. 7e, f). While the longitudinal force (Fig. 7e) and the propulsion enhancement (Fig. 7g) are both decreasing with the reduction of $\tilde{R} = R_{cap}/R_p$, the absolute value of the radial force is increasing until it reaches $|F_{SP,y}|$ (note that the mismatch between $F_{SP,y,d=10nm}$ and $F_{SP,y,d=20nm}$ at $\tilde{R} \to 0$ is an artefact due to meshing). This result is consistent with the

transverse trapping of Janus particles being effectively stronger than expected from simulations. Furthermore, we find a good agreement between the calculated and measured values of the propulsion enhancement at $\tilde{R} \approx 93.4\%$ (Fig. 7g). This rather crude model allows us to obtain realistic estimates for the temperature increase and the light-induced propulsion without the need to scale down optical forces or include thermophoresis. Of course, for better accuracy of the the-oretical analysis, one should consider directly incorporating nanoscale irregularities of the gold film into the model, and accounting for thermal[53] and electrostatic[54] effects relevant for hydrodynamic beha-vior of the system. The resultant model would have to be verified by direct visualizations of the fluid motion using tracer nanoparticles[15].

In this work we have reported on the optical manipulation of composite metallo-dielectric Janus particles (silica microspheres half-coated with a thin layer of gold) in the evanescent field of an optical nanofiber. Our experiments demonstrate that such particles are stea-dily propelled along the mode propagation direction, with speeds exceeding those typically measured for uncoated silica beads of the same size. Notably, we registered a speed enhancement factor of $2.56 \pm 0.13$ for $0.59$-μm-thick nanofibers and $3$-μm particles with 20-nm-thick gold coating. This enhancement depends on geometric parameters (fiber radius and the coating thickness), but does not depend on the optical power; hence, Janus particles in the evanescent field exhibit largely the same light-induced dynamics as standard dielectric beads: (i) steady propulsion with a speed proportional to the optical power and (ii) negligibly small thermal effects (at least as far as the propulsion is concerned). For the above parameters, the propul-sion speed was $2.13 \pm 0.11$ μm per second per 1 mW, equivalent to 71 body length per second for the transmitted power of 100 mW. This is 2.84 times faster than the optical propulsion of similar Janus particles in free-space[23].

Comparing the experimental and theoretical results, we link the propulsion enhancement to the orientation of the gold cap, which is consistently recorded to be facing away from the nanofiber and toward the direction of motion, effectively acting as an optical "sail". The cap's stable orientation, $\alpha_0$, is found to be independent of the optical power.

This validates our assumption that the total torque on a Janus particle near a nanofiber can be attributed to optical forces alone. Indeed, our model (Eq. (1)) accounts for the redirection of light at the particle's surface (a motion-independent optical torque) and for the dynamic friction originating from the radial optical force pressing the moving particle to the nanofiber, both impacts being proportional to the photon flux. Theoretical $\alpha_O$ values, calculated with the once-adjusted friction coefficient, are in good agreement with those directly measured by imaging of stably propelled Janus particles.

However, the propulsion speeds calculated for these $\alpha_O$ values turned out to be significantly higher (by about 2.8 and 3.6 times for 10- and 20-nm-thick coatings, respectively) than the measured speeds, whereas simulations for uncoated silica beads were quite accurate. In addition, our thermal model predicts unrealistically high temperature increase which would cause water to boil near a Janus particle at optical power about 20 mW or higher. These results suggest that nanoscale irregularities of the gold coating play a much more important role in optomechanics of Janus particles than we expected. In particular, oblique and nonuniform deposition of gold near the edges of the gold cap might severely affect its crystallinity, plasmonic properties, and ultimately the absorption which is directly linked to the light-induced heating. We confirmed this conclusion numerically by removing a part of the gold cap near its edges. Indeed, if the effective base radius of the cap is reduced by as little as 6.6% (while keeping $\alpha_O$ and the thickness constant), the simulated propulsion enhancement for Janus particles agrees well with the experimental data, and the temperature does not exceed the boiling point of water. Hopefully, realistic modeling of local irregularities in metallo-dielectric particles will become possible with future development of computational resources.

Aside from the efficient propulsion, we have found that Janus particles exhibit strong optical trapping in the polarization plane, albeit with a lower trap stiffness (down to 61% for 10-nm-thick coating and 30% for 20-nm-thick one) compared to that of uncoated silica beads. Overall, the optical nanofiber is an effective tool for precise optical manipulation of Janus particles and their applications as micro- or nanoscale actuators or transporters. We envisage further development of plasmonic[55] and waveguide-based technologies with Janus particles, in particular contributing to the prospective "rail"-style microfluidic designs[56,57].

## Methods

### Simulations of optical forces and torques

To model optomechanics of a Janus particle, we first find the distributions of the surrounding electric (**E**) and magnetic (**B**) fields by solving the scattering problem numerically via the finite element method (COMSOL 5.5 Multiphysics package). These fields are then used to define the time-independent Maxwell's stress tensor, $\overleftrightarrow{T}$, with elements $T_{ij} = \varepsilon_m(E_i E_j - \frac{1}{2}|\mathbf{E}|^2 \delta_{ij}) + \frac{1}{\mu_0}(B_i B_j - \frac{1}{2}|\mathbf{B}|^2 \delta_{ij})$, where $\varepsilon_m$ is the permittivity of the medium surrounding the particle, $\mu_0$ is the vacuum permeability, and $\delta_{ij}$ is the Kronecker delta. The total optical force and torque are found by numerical integration of $\overleftrightarrow{T}$ over the surface enclosing the particle[58]

$$\mathbf{F} = \langle \oint_S \hat{n} \cdot \overleftrightarrow{T}\, dS \rangle_t \tag{4}$$

$$\mathbf{N} = - \langle \oint_S \hat{n} \cdot \left( \overleftrightarrow{T} \times \mathbf{r} \right) dS \rangle_t \tag{5}$$

where $\langle \ldots \rangle_t$ represents a time average, $\hat{n}$ is the outwardly directed normal unit vector, and **r** is the radius vector.

Our model is outlined in Fig. 2a. For all calculations, the total propagating power of the guided light was normalized to 1 mW. The JPs are modeled by a silica sphere with concentric half-dome caps of titanium and gold. The radial force, $\mathbf{F}_y$, originates from the transient

polarization of the trapped particle in a non-uniform field. The longitudinal force, $\mathbf{F}_z$, is responsible for propelling the particles along the fiber. We set the enclosing surface for integration to be positioned at 5 nm from the outermost edge of the particle. The refractive indices of silica structures and the surrounding water were taken as 1.45 and 1.33, respectively. Optical properties of metals were defined using the Brendel-Bormann model[46]. We use the simulation domain of $(x \times y \times z) = (5\,\mu m \times 5\,\mu m \times 15\,\mu m)$. Scattering condition layers are applied at all exterior boundaries to avoid back-reflection of the outgoing wave. The mesh is free tetrahedral; the minimum and maximum mesh element sizes for metallic caps are 5 nm and 50 nm, respectively. For silica objects (microspheres and ONFs), the minimum and maximum mesh element sizes are 20 nm and 105 nm, respectively. The model requires 0.75 TB of memory to perform the calculation. The ONF radii and JP orientations are studied using parametric sweep. Calculation of the 25 data points for each graph takes 30 hours of machine time on the OIST Graduate University computing cluster.

### Particle preparation

Silica-gold JPs were prepared following the evaporation coating method[29]. First, a loosely packed monolayer of silica microspheres ($2R_p = 3.13 \pm 0.20\,\mu m$ from Bangs Laboratories, Inc.) was formed on a glass substrate by drop-casting them in an ethanol suspension and drying under ambient conditions. Second, the substrate was loaded into an electron beam evaporator (MEB550S2-HV by PLASSYS-BESTEK) and sequentially coated with a 5-nm-thick adhesion layer of titanium and a 10- or 20-nm-thick layer of gold. Finally, JPs were detached from the substrate by sonication for 10 minutes in Milli-Q water. This stock suspension of particles was further diluted by Milli-Q water to achieve a convenient concentration (about 5 particles in the full field-of-view, FoV) for the optomechanical experiments. All solutions of SPs and JPs had pH = 8.2 ± 0.2. We also measured the zeta-potential (Zetasizer Nano ZS by Malvern Panalytical Ltd.) of JP solutions to be − 25.1 ± 5.9 mV for 10-nm-thick gold coating and − 27.4 ± 3.9 mV for 20-nm-thick one. Under these conditions, the solutions were electrostatically stable and the particles did not aggregate or stick to the fiber or the glass slides. As a test, we prepared a pH = 4.6 solution (sodium acetate trihydrate and acetic acid in water) and found that it was unstable, meaning that JPs stuck to the glass. Some could be lifted by increasing the power in the optical tweezers to ~ 20 mW (enough to produce cavitation at the gold cap), but then the particles stuck to the fiber once brought in contact with it.

### Optical nanofiber sample fabrication

The ONFs were fabricated via controlled heating and pulling of step-index single-mode optical fibers (SM980G80 by Thorlabs, Inc.) aiming at an adiabatic exponential taper profile[59]. Adiabaticity was verified by in situ monitoring of the transmission of a probe laser beam (1.064 μm wavelength) coupled to one of the fiber pigtails; only the samples having > 98% transmission were used in the optomechanical experiments. The ONF waist regions had a nearly constant (up to ± 5 nm) radius within at least a 2-mm-long stretch, as measured by SEM (FEI Quanta 250). A freshly pulled ONF was suspended over a 0.15-mm-thick glass cover slip on pieces of 0.08-mm-thick tape and secured by UV adhesive glue (NOA81 by Norland Products, Inc.). After depositing about 0.3 mL of particle suspension in Milli-Q water onto the ONF waist and taper regions, we covered them with a second glass cover slip, supported on 1-mm-thick polymer spacers. The sample had its sides open to the atmosphere, thus all the measurements on each were performed in one session, in order to avoid detrimental effects of water evaporation.

### Experimental setup

Samples were mounted on a three-axis mechanical stage (MAX313D by Thorlabs, Inc.). Beams 1-3 (Fig. 3a and Supplementary Figure 6) were

collimated, non-interfering, linearly polarized Gaussian beams from the same laser source (Ventus by Laser Quantum Ltd.). Beams 1 and 2 were coupled to the fiber pigtails of the sample. We used the guided modes polarized in the vertical ($yz$) or horizontal ($xz$) planes. To this end, we set the input polarization state in each of the beams 1 and 2 to vertical (**V**, along $y$) or horizontal (**H**, along $x$) by means of half-wave plates and performed a partial polarization compensation procedure by mapping $H \rightarrow H$ at the ONF waist[60]. The procedure relies on the independent adjustment of two quarter-wave plates in each compensator, PC1 and PC2, while imaging the laser light scattered from the waist[61]. The mapping is complete when the sum of the pixel brightness over the image reaches its absolute maximum.

The optical tweezers for trapping and delivering individual particles to an ONF was realized by tightly focusing beam 3 via a water-immersion objective lens (Zeiss Plan-Apochromat, $63 \times /1.00w$). Transmission imaging of the sample under Köhler illumination was performed through the same objective, which projected the image (Fig. 3b) onto a video camera (DCC3240C by Thorlabs, Inc.) through a tube lens with 75-mm-long focal distance. To ensure correct timing of the camera operated by the native Thorlabs software, we manually cropped the FoV to about 1/6 of its full size (fitting the fiber and the particle) and set the pixel clock to be under 20 MHz, which reliably gave a frame rate of $91 \pm 1.2$ fps.

The optical power in the tweezers was measured in real time by a calibrated photodetector (PDA10A2 by Thorlabs, Inc.) capturing a small portion of beam 3 reflected off a polarizing beamsplitter. To estimate the optical power at the ONF waist, we deflected a few percent of the transmitted light from the driving beam 1 by a glass wedge to another calibrated photodetector. Although we cannot measure the power at the waist directly, this transmission-based measurement is expected to give an accurate estimate, because every sample ONF had over 98% transmittance as-fabricated, and we expect most of the losses to occur due to radiation modes launched from the down-taper region. The power was measured just before bringing a particle into contact with the ONF and checked after releasing the particle at the end of a propulsion cycle. We found the power to be stable within $\pm 2\%$. Real-time measurements of the ONF power were inaccurate, because the transmitted power was seen to vary in about $\pm 20\%$ range due to scattering by the particle. In our experiments, the transmitted ONF power varies from 20 mW to 100 mW in steps of 10 mW. While working with each sample, we checked the efficiency of fiber coupling for beam 1 (about 60% for an average sample) at least once per hour, as a control for the ONF cleanliness and the polarization accuracy.

### Measurement procedure

Before starting the propulsion experiments, we first determined the thinnest and most uniform part of the ONF waist by near-field opto-mechanical probing. For this purpose, we selected a SP with the optical tweezers, brought it to the ONF, unblocked beam 1 so the particle would be trapped near the ONF, and switched off the tweezers by blocking beam 3. Then we observed the particle's motion and moved the sample along $z$, until we found the position where $V_z$ was the highest and the most uniform across the FoV.

A typical cycle of the propulsion measurements consisted of these steps: (i) unblock beam 3 to form an optical trap around the center of the FoV, (ii) trap a JP and bring it to the ONF, (iii) unblock beams 1, 2 and block beam 3 to transfer the particle into the evanescent field trap, (iv) block beam 1 to propel the particle toward $-z$; (v) after it has passed the left edge of the FoV, unblock beam 1, block beam 2, and record the particle's propulsion all the way to the right edge of the FoV, (vi) block all beams and repeat steps (i)-(v) for a SP. A complete data set from a single sample contained at least 5 JPs and 2 SPs (the latter had much more reproducible behavior) for each power value. Once the data were collected, successful samples were left to dry, sputter-coated with a

few nanometers of Pt-Pd and imaged by SEM. The particle's position was measured by analyzing video recordings in MATLAB (Supplementary Note 3).

## Data availability

The data that support the findings of this study are provided as a collection of Excel spreadsheets in the Supplementary Information/Source Data file. In addition, these data and the corresponding source video files have been deposited in the public repository Figshare under accession code https://doi.org/10.6084/m9.figshare.21993455. The measured gold cap orientation angles for JPs (Fig. 3c, Fig. 5c) are provided in Supplementary Figure 7. Source data are provided with this paper.

## Code availability

The COMSOL model used for numerical simulations in this study and the MATLAB codes used to process the source videos in order to generate and plot the experimental data have been deposited in the public repository Figshare under accession code https://doi.org/10.6084/m9.figshare.21993455. The function used for particle tracking is provided in Supplementary Note 3.

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

## Acknowledgements

We thank K. Karlsson, M. Ozer, the Engineering Section, and the Scientific Computing and Data Analysis Section of Okinawa Institute of Science and Technology Graduate University (OIST) for technical assistance. We also thank M. Dindo from the Protein Engineering and Evolution Unit at OIST for measuring the pH of our solutions. This work was funded by OIST and the Japan Society for the Promotion of Science (JSPS) KAKENHI (Grant-in-Aid for JSPS Fellows) 18F18367. G.T. was supported by a JSPS International Research Fellowship (Standard) P18367.

## Author contributions

S.N.C. initiated and supervised the project. V.G.T. and C.L.E. performed simulations. G.T., V.G.T., and I.S. performed fabrication and SEM-imaging of Janus particles. G.T. and C.L.E. conducted the experiments and analyzed the data. G.T. built the optical setup, prepared graphic materials, and wrote the manuscript with input from all authors.

## Competing interests

The authors declare no competing interests.
