## [Peer Review File · Nature Communications]

Evanescent field trapping and propulsion of Janus particles along optical nanofibersEditorial Note: Parts of this Peer Review File have been redacted as indicated to remove third-party material where no permission to publish could be obtained.

REVIEWER COMMENTS

Reviewer #1 (Remarks to the Author):

This manuscript describes the experimental demonstration of optical trapping and propulsion of Janus particles using optical nanofibers. The authors employed the optical forces of the evanescent field formed around the optical nanofiber with the 700-nm diameter. They realized trapping of silica microspheres half-coated with nanometer-thick layer of gold by the gradient force and succeeded in transporting them along the nanofiber. Furthermore, orientation of the gold-coated silica spheres is finely controlled by the optical force. Manipulation and precise positioning of various functional Janus particles is one of the challenging research topics, therefore, the proposed technique and its experimental demonstrations will interest the readers of Nature Communications. However, it is difficult to recommend the present manuscript for publication because of the following reasons.

1. Trapping and manipulation of Janus particles using optical forces of focused laser beams have been reported in several papers. In addition, several research groups and the present authors have reported on optical trapping and transportation of micro- and nano-sized particles using the evanescent field of the optical nanofibers, as mentioned in the manuscript. Therefore, the originality and novelty of this manuscript should be in the optical control of the orientations of the Janus particles. However, there is no discussion on the mechanism of the orientation control based on theories and simulations of the optical forces and optical torques, which is the critical problem in this manuscript.

2. The size of the particles used in the experiments is 3.13 ± 0.2 μm . Different particles were used for individual measurements. How large deviation of the optical force is caused by the size variation of ± 0.2 μm ? The ~ 3 - μm sphere has whispering gallery resonances, which may cause the drastic change in the optical force depending on the particles size. The error bars are shown in Figs. 4 and 5, however, they are deviations of the propulsion speeds observed at different positions on the nanofibers not the deviation for different particles.

3. The simplified two-dimensional model is used for numerical simulations. I think that precise three-dimensional simulation is indispensable for quantitative discussion on optical forces.

Reviewer #2 (Remarks to the Author):

This paper reports on the experimental observation of trapping and propulsion of silica or metal-capped silica particle along an optical nano-fiber.

Regarding the underlying mechanisms, from their numerical simulations of wave propagation, the authors conclude that the trapping is due to the tweezers potential whereas driving results from symmetry breaking due to the laser light propagating in the fiber. Yet comparison of the trapping strength of silica and JP suggests that this picture is not fully consistent, and thermal forces could play a role.

This is a nice work that could be of broad interest, given the conceptually simple fiber-particle-laser setup, the precise control of the particle, and the high speeds obtained.

The paper is rather well written, results are well presented.

- Originality. The second-but-last paragraph of the introduction states "There have been numerous works on optical manipulation via evanescent fields..." In the last paragraph the authors give the detail of their approach. In order to clarify to which extent the present setup and experimental procedure are original, it would be helpful to state which aspects parallel previous work [32-45] and which go beyond.

- Mechanical model. The authors should give more details on their simulations. In particular, they

should discuss possible hydrodynamic effects, even if their model does not account for fluid motion.

It would be interesting if the authors could calculate the torque exerted on the particle, in addition to the vertical and longitudinal forces. The particle's drag coefficient stems mostly from the area close to the fiber. On the other hand, the longitudinal force is probably strongest in the particle volume close to the fiber. Thus there could be a significant torque, resulting in rolling motion of silica spheres and inclination of the JP.

More generally, hydrodynamic coupling affects particle motion driven by optical forces yet has little effect on thermophoresis. Perhaps the authors could include hydrodynamic aspects in the comparison of JP/silica beads and optical driving/thermophoresis.

Both the radial particle-fiber distance and the longitudinal velocity scatter significantly. It would be interesting to evaluate their correlation. From the evanescent wave mechanism one would expect that the velocity is larger at small distances, whereas the lubrication drag coefficient should result in the opposite behavior.

- Physical chemistry parameters. The authors mention thermophoresis as a possible source for the observed inconsistency of their model and data. Thermal forces depend on properties of the liquid and of the surfaces. The authors mention that there is no effect in heavy water, is there any explanation for that? Have they tried pure water at pH 4.6 instead of milliQ water? If available more information on the particle and fiber surfaces could be interesting (zeta potential, functionalization...)

Reviewer #3 (Remarks to the Author):

The manuscript by Esporlas and co-workers describes simulations and experiments of optical trapping and propulsion of Janus spheres (Au coated silica) along nanofibers in water. The main conclusion is that the Janus particles exhibit slightly higher propulsion speed and trap stiffness compared to silica spheres of the same size. Though interesting, the results are not very surprising given the well known higher polarizability/reflectivity of Au films, leading to enhanced radiation pressure effects and gradient forces.

I'm also not convinced that the methodology for trapping and propulsion of Janus particles used here has much practical relevance since the setup and requirements are obviously very delicate and complicated (the methodology also seems to be the same as used in several previous publications from the Chormaic group, so it's difficult to argue for any conceptual advancements from the point of view of experimental design).

Specific comments:

1) My understanding is that the conclusions are based on measurements of very few particles. In Fig. 3 there are only data for 4 particles shown, two Janus with different coating thickness and two silica of nominally identical size. Since the latter show significantly different speeds, one wonders what the statistical significance of the Janus results are? In these kind of "single particle" experiments, one obviously needs to demonstrate statistical significance - I would say that 5 particles of each kind is a minimum!

2) The simulations are based on a Maxwell stress tensor approach realized in Comsol. The Maxwell stress tensor can also be used to calculate optical torques. By doing that, it should be quite straightforward to find the equilibrium angle α in Fig. 1 (i.e. when the torque around the x-axis vanishes).

The paper is well written and the results and conclusions are scientifically sound in general (except for the question on statistical relevance). However, I don't think that the results are significant enough or of high enough general interest for publication in Nature Communications. I therefore recommend resubmission to a more specialized optics journal or transfer to Scientific Reports.

Manuscript ID: NCOMMS-21-32539

“Evanescent field trapping and propulsion of Janus particles along optical nanofibers”

by Georgiy Tkachenko, Viet Giang Truong, Cindy Liza Esparlas, Isha Sanskriti, and Síle Nic Chormaic.

Response to the reviewers' reports

Reviewer 1

This manuscript describes the experimental demonstration of optical trapping and propulsion of Janus particles using optical nanofibers. The authors employed the optical forces of the evanescent field formed around the optical nanofiber with the 700-nm diameter. They realized trapping of silica microspheres half-coated with nanometer-thick layer of gold by the gradient force and succeeded in transporting them along the nanofiber. Furthermore, orientation of the gold-coated silica spheres is finely controlled by the optical force. Manipulation and precise positioning of various functional Janus particles is one of the challenging research topics, therefore, the proposed technique and its experimental demonstrations will interest the readers of Nature Communications.

Answer: We thank the reviewer for acknowledging the quality of our work and for the positive recommendation regarding its suitability for Nature Communications.

However, it is difficult to recommend the present manuscript for publication because of the following reasons.

1. Trapping and manipulation of Janus particles using optical forces of focused laser beams have been reported in several papers. In addition, several research groups and the present authors have reported on optical trapping and transportation of micro- and nano-sized particles using the evanescent field of the optical nanofibers, as mentioned in the manuscript. Therefore, the originality and novelty of this manuscript should be in the optical control of the orientations of the Janus particles. However, there is no discussion on the mechanism of the orientation control based on theories and simulations of the optical forces and optical torques, which is the critical problem in this manuscript.

Answer: We agree that [free-space] optical manipulation of Janus particles and [near-field] manipulation of micro- or nanoparticles with optical nanofibers are both well-known. However, in this work we investigate a combination; a Janus particle manipulated by evanescent fields. This approach is both new and effective, because it allows one to manipulate metallo-dielectric particles almost as if they were all-dielectric, that is free from the drawbacks and limitations of heating. Moreover, the physical problem under study is far from trivial, its solution has required careful experiments and numerical simulations at the limit of state-of-the-art computational resources. Therefore, we are certain that our work will be of interest to a broad scientific audience.

Admittedly, the original manuscript did not give a proper discussion on the particle's orientation in the evanescent field. This flaw has been amended by our new theoretical and experimental analyses presented in the revised version.

2. The size of the particles used in the experiments is 3.13 ± 0.2 μm . Different particles were used for individual measurements. How large deviation of the optical force is caused by the size variation of ± 0.2 μm ? The ~ 3 - μm sphere has whispering gallery resonances, which may cause

the drastic change in the optical force depending on the particles size. The error bars are shown in Figs. 4 and 5, however, they are deviations of the propulsion speeds observed at different positions on the nanofibers not the deviation for different particles.

Answer: Our original set of experimental data did not have a statistical relevance, precisely because we used a single Janus particle for each sample, and the particles could have slightly different geometries. In the revised study, we have analyzed at least 5 different Janus particles and 3 silica particles for each nanofiber sample and each optical power value. We found that standard deviation ranges for Janus particles are much larger (up to an order of magnitude) compared to those for silica. This suggests that the size variations of the core particle did not have a major contribution to the errors, in contrast to random irregularities of the gold coating (in particular, the nanoscale kinks in the gold cap's edge where the drop-casted silica particles were touching each other during the e-beam evaporation coating). To test this conclusion, we have investigated the impact of size variation by $\pm 0.2 \mu\text{m}$ theoretically (see the below graphs, which are now part of Supplementary Figure 3). We found that the balance orientation of the gold cap is almost unchanged, and the corresponding values of the longitudinal optical force (and the propulsion enhancement) vary by less than 2%. This uncertainty is negligible in comparison with the standard deviation ranges obtained for Janus particles in our experiments.

Regarding the whispering gallery resonances, we agree that they may influence the electromagnetic field pattern around the particle. However, with the particle radius being comparable to the working wavelength of light, the resonances will have Q-factors close to 1, and even worse for Janus particles where the gold coating will be a source of strong scattering losses. In any case, our numerical model solves the scattering problem for the given geometry and outputs the field distribution which must be physically correct (up to some limitations due to the finite mesh, of course).

3. The simplified two-dimensional model is used for numerical simulations. I think that precise three-dimensional simulation is indispensable for quantitative discussion on optical forces.

Answer: We agree. Indeed, as written in the revised manuscript

a simplified 2D model might seem sufficient for simulation of this optomechanical system. However, such a model treats the gold coating as a half-cylinder instead of a half-dome cap, thereby leading to a significant deviation from the real physical picture. Consequently, we have to use a full 3D numerical model, with its cross-section through the symmetry plane sketched in Fig. 2a.

Reviewer 2

This paper reports on the experimental observation of trapping and propulsion of silica or metal-capped silica particle along an optical nano-fiber.

Regarding the underlying mechanisms, from their numerical simulations of wave propagation, the authors conclude that the trapping is due to the tweezers potential whereas driving results from symmetry breaking due to the laser light propagating in the fiber. Yet comparison of the trapping strength of silica and JP suggests that this picture is not fully consistent, and thermal forces could play a role.

This is a nice work that could be of broad interest, given the conceptually simple fiber-particle-laser setup, the precise control of the particle, and the high speeds obtained.

The paper is rather well written, results are well presented.

Answer: We thank the reviewer for the positive feedback on the quality and scope of our work. We hope that both have been improved in the revised version.

- Originality. The second-but-last paragraph of the introduction states "There have been numerous works on optical manipulation via evanescent fields..." In the last paragraph the authors give the detail of their approach. In order to clarify to which extent the present setup and experimental procedure are original, it would be helpful to state which aspects parallel previous work [32-45] and which go beyond.

Answer: The study [32] concerns evanescent fields produced by total internal reflection and [33] reports on the use of planar waveguides, which have not become a major tool for particle manipulation, in contrast to optical nanofibers. References [34-45] all use nanofibers basically in the same way. However, they have never been applied to the manipulation of a composite metallo-dielectric object, which is the purpose of our study. To emphasize the originality of this problem, we updated the last sentence of that paragraph as follows: "... optical nanofiber (ONF), which proved to be very effective in producing propulsion [34-38], binding [36, 39-41], rotation [42], detection [43], and sorting [44, 45] of various kinds of micro- and nanoparticles, but never metallo-dielectric ones".

- Mechanical model. The authors should give more details on their simulations. In particular, they should discuss possible hydrodynamic effects, even if their model does not account for fluid motion.

It would be interesting if the authors could calculate the torque exerted on the particle, in addition to the vertical and longitudinal forces. The particle's drag coefficient stems mostly from the area close to the fiber. On the other hand, the longitudinal force is probably strongest in the particle volume close to the fiber. Thus there could be a significant torque, resulting in rolling motion of silica spheres and inclination of the JP.?

Answer: We are grateful to the reviewer for this suggestion, which we took on board. Once we changed the optomechanical model from 2D to 3D and implemented calculations of the optical torque, we found that the torque as a function of the particle's orientation never crossed zero. Hence, a particle subjected to this torque alone would be rotating continuously and this result contradicts our observations. Importantly, the friction force at the particle-nanofiber contact

leads to another torque component which is antiparallel to the optical torque. As a result, the total torque on a Janus particle moving along the nanofiber does go through zero with a negative slope, thereby creating a restorative torque at the corresponding orientation angle. In the revised manuscript, we study this angle both theoretically and experimentally as a function of the gold coating thickness and the fiber radius, which has shown to be a crucial geometric parameter.

More generally, hydrodynamic coupling affects particle motion driven by optical forces yet has little effect on thermophoresis. Perhaps the authors could include hydrodynamic aspects in the comparison of JP/silica beads and optical driving/thermophoresis.

Answer: True, it would be interesting to study thermal effects in the JP/ONF system. However, our goal in this work was to demonstrate that JPs can be manipulated primarily by optical forces, with ONF being an effective tool for such manipulation. Indeed, the propulsion speeds for JPs were found to be proportional to the optical power and nearly constant throughout each recorded track. These results indicate that thermophoresis could be safely neglected as far as light-induced propulsion was concerned. However, thermodynamical aspects apparently were responsible for somewhat stronger transverse localization of JPs compared to the theory predictions, see the first paragraph of “Discussion”. We leave these aspects for the future.

Both the radial particle-fiber distance and the longitudinal velocity scatter significantly. It would be interesting to evaluate their correlation. From the evanescent wave mechanism one would expect that the velocity is larger at small distances, whereas the lubrication drag coefficient should result in the opposite behavior.

Answer: Following the experimental study [50], where 1-5 μm silica beads were manipulated by an evanescent focal spot produced by total internal reflection, and our own experiment with 3 μm silica beads near single-mode nanofibers [42], we consider $\delta = 10 \text{ nm}$ as a good estimate for the distance between a glass surface and a glass microsphere pressed to it by a gradient optical force. This distance is due to the surface roughness and perhaps the hydrodynamic lift [47], and we cannot measure it directly. Following the reviewer’s suggestion, we have simulated optical forces and torques for δ ranging from 10 nm to 150 nm, see the below graph which is now part of Supplementary Figure 3. We found that the balance orientation of the gold cap is almost unchanged, and the corresponding values of the longitudinal optical force (and the propulsion enhancement) vary by less than 2%. This uncertainty is negligible in comparison with the standard deviation ranges obtained for Janus particles in our experiments.

- Physical chemistry parameters. The authors mention thermophoresis as a possible source for the observed inconsistency of their model and data. Thermal forces depend on properties of the

liquid and of the surfaces. The authors mention that there is no effect in heavy water, is there any explanation for that? Have they tried pure water at pH 4.6 instead of milliQ water? If available more information on the particle and fiber surfaces could be interesting (zeta potential, functionalization...)

Answer: Manipulation of Janus particles in heavy water with optical tweezers was very challenging. Although particles could be lifted from the bottom glass slide and pushed upwards quite easily, they promptly escaped when we tried to move them laterally or when the height exceeded a few tens of micrometers, so they could not be brought to the nanofiber elevated by about 120 μm . Since the refractive indices of H_2O and D_2O differ by less than 0.5%, we do not expect significant spherical aberrations for our water-immersion objective. Therefore, we consider thermal effects as a more likely explanation for this behavior of JPs in optical tweezers.

Nevertheless, we have succeeded to deliver a JP to a nanofiber which was placed at about 20 μm from the bottom glass slide (this can be achieved when the tapered fiber is glued on the slide without spacers, so that the glue could pull and slightly bend the taper regions by the surface tension). The behavior of the particle (trapping, propulsion, balance orientation) in D_2O was almost the same as that in H_2O . We could not perform a statistically relevant study, however, because with the fiber in this configuration other particles were spontaneously “captured” by the evanescent field, usually outside the field-of-view of our imaging system.

To answer the question about the manipulation in a low-pH fluid (which could not be pure water as it has pH of 7), we prepared a 0.2M water solution of sodium acetate with the desired pH of 4.6. Once dispersed in this liquid, both Janus and silica particles firmly attached to the bottom glass slide due to electrostatic attraction. Silica particles could not be moved at all by optical tweezers, but we managed to detach some Janus particles by creating cavitation around gold caps exposed to the tweezer beam at >20 mW of power. However, when delivered to the nanofiber, particles became permanently stuck on it, so we could not perform any propulsion experiments. We checked the pH of our particle solutions in pure water, the result was 8.2 ± 0.2 .

Following the reviewer’s suggestion, we have also measured the zeta-potential of JP solutions and found (-25.1 ± 5.9) mV for 10-nm-thick gold coating and (-27.4 ± 3.9) mV for 20-nm-thick one. According to the below graph (from the manual of our Zetasizer Nano ZS by Malvern Panalytical Ltd.), such solutions must be electrostatically stable, meaning that the particles should not aggregate or stick to the glass, as we indeed observed experimentally.

Redacted

The above information has been included in the “Methods” section of the revised manuscript.

Reviewer 3

The manuscript by Esporlas and co-workers describes simulations and experiments of optical trapping and propulsion of Janus spheres (Au coated silica) along nanofibers in water. The main conclusion is that the Janus particles exhibit slightly higher propulsion speed and trap stiffness compared to silica spheres of the same size. Though interesting, the results are not very surprising given the well known higher polarizability/reflectivity of Au films, leading to enhanced radiation pressure effects and gradient forces.

Answer: We thank the reviewer for acknowledging that our results are interesting. Certainly, the propulsion enhancement due to gold is not very surprising. However, the ability to stably trap metallo-dielectric particles and manipulate them essentially as if they were all-dielectric is quite surprising, giving the available research on Janus particles. Moreover, an understanding of the underlying physics requires both numerical simulations and accurate experiments, hence this nontrivial work should be of interest to a broad scientific audience.

I'm also not convinced that the methodology for trapping and propulsion of Janus particles used here has much practical relevance since the setup and requirements are obviously very delicate and complicated (the methodology also seems to be the same as used in several previous publications from the Nic Chormaic group, so it's difficult to argue for any conceptual advancements from the point of view of experimental design).

Answer: This methodology is standard practice for all experiments with micro- and nanoparticles near optical nanofibers. Researchers who wish to perform such experiments in a clean way need to isolate the nanofiber from random interactions with colloidal particles and instead to pick and/or image the target particles using optical tweezers. This work did not pursue any improvements in the optical manipulation or characterization methods. Instead, the established methods were applied to investigate a novel approach to the manipulation of Janus particles, with the focus on theoretical and experimental analyses of the particles' behavior. As for the practical relevance of this approach, its implementation can be very simple, for instance through a microfluidic channel with a nanofiber or a planar waveguide and (optional) microlenses for particle loading. Here we study the scientific concept, not its commercial potential.

Specific comments:

1) My understanding is that the conclusions are based on measurements of very few particles. In Fig. 3 there are only data for 4 particles shown, two Janus with different coating thickness and two silica of nominally identical size. Since the latter show significantly different speeds, one wonder what the statistical significance of the Janus results are? In these kind of "single particle" experiments, one obviously needs to demonstrate statistical significance - I would say that 5 particles of each kind is a minimum!

Answer: We thank the reviewer for this valuable comment. Indeed, previously we analyzed only one Janus particle per nanofiber sample. Since the particles (and the fibers, as it turns out) of the same kind could have somewhat different geometries, those results were quite poor in terms of statistical relevance. In the revised version, we have analyzed at least 5 different Janus particles and 3 silica particles for each nanofiber sample and each optical power value. A smaller number of silica particles was sufficient since we found that standard deviation ranges for Janus particles are much larger (up to an order of magnitude) compared to those for silica. In contrast to our previous approach where the error bars were obtained from variations of the particle's speed within each track, now the statistics is made of the mean speed values of a selection of particles measured for the same optical and geometrical conditions. The balance orientation angle of the

gold cap has also been measured over multiple particles, and the imaging resolution has been improved.

2) The simulations are based on a Maxwell stress tensor approach realized in Comsol. The Maxwell stress tensor can also be used to calculate optical torques. By doing that, it should be quite straightforward to find the equilibrium angle θ in Fig. 1 (i.e. when the torque around the x-axis vanishes).

Answer: We agree, and these calculations have been implemented.

The paper is well written and the results and conclusions are scientifically sound in general (except for the question on statistical relevance). However, I don't think that the results are significant enough or of high enough general interest for publication in Nature Communications. I therefore recommend resubmission to a more specialized optics journal or transfer to Scientific Reports.

Answer: Again, we appreciate the positive feedback on our work. Since we have amended the issue with statistical relevance and all other flaws spotted by the reviewers, we believe that the revised manuscript and the new supplementary information do meet the standards of Nature Communications.

REVIEWER COMMENTS

Reviewer #1 (Remarks to the Author):

In the revised manuscript, the authors used a full 3-D numerical model and quantitatively and statistically discussed on optical forces and torques exerted on silica microspheres half-coated with nanometer-thick layer of gold, according to the reviewer's comments. Thus, I would recommend that the manuscript can be accepted for publication. However, I think the author should revise the manuscript regarding the following points before publication.

1. The authors determined the friction coefficient (μ) to be 0.25 by fitting the experimental data and the theoretical analysis. Is this value reasonable? I think that the friction coefficient can be estimated from known or reported physical parameters.
2. The separation distance (δ) between surfaces of a particle and an optical nanofiber is the important parameter, which affects the friction coefficient. However, there is no discussion on this parameter. I think the separation distance depends on the temperature.
3. The authors can estimate the temperature distribution under the present experimental conditions by the COMSOL software. I think that the temperature distribution might be different between silica particles and Janus particles and be dependent of the size and gold-layer-thickness. It is necessary to discuss on the temperature dependence.
4. The thermal fluctuation is also important for analysis of the particle motion in liquid. The temporal variation of the particle position affects the separation distance (δ) and the friction coefficient (μ) and then the orientation angle (α). I think the author should discuss on this issue.

Reviewer #2 (Remarks to the Author):

The revised version has significantly improved, in the response to the previous reports the authors have done additional experiments and clarified various issues. The authors have made quite an effort to confirm their model for the motion along the fiber. Now this is a sound and rather complete work on optical manipulation of JP.

The experimental setup is rather well known from previous studies of colloidal particles, original results concern characterizing and understanding particle motility. The overall picture is rather convincing, even if important discrepancies persist regarding the torques and directional enhancement factors.

Though this is a nice paper, I am not entirely convinced it warrants publication in Nature Communications.

Manuscript ID: NCOMMS-21-32539A

“Evanescent field trapping and propulsion of Janus particles along optical nanofibers”

by Georgiy Tkachenko, Viet Giang Truong, Cindy Liza Esparlas, Isha Sanskriti, and Síle Nic Chormaic.

Response to the reviewers' reports on the revised version

Reviewer 1

In the revised manuscript, the authors used a full 3-D numerical model and quantitatively and statistically discussed on optical forces and torques exerted on silica microspheres half-coated with nanometer-thick layer of gold, according to the reviewer's comments. Thus, I would recommend that the manuscript can be accepted for publication.

Answer: We thank the reviewer for the positive evaluation of our work.

However, I think the author should revise the manuscript regarding the following points before publication.

1. The authors determined the friction coefficient (μ) to be 0.25 by fitting the experimental data and the theoretical analysis. Is this value reasonable? I think that the friction coefficient can be estimated from known or reported physical parameters.

Answer: Friction coefficients of 0.1-0.6 are typical for lubricated and greasy glass surfaces (relevant to our case where water acts as a lubricant), according to online resources (see for instance https://www.engineeringtoolbox.com/friction-coefficients-d_778.html, https://www.engineersedge.com/coefficients_of_friction.htm). The value of 0.21 was measured for centimeter-scale lubricated pieces of glass [K. H. Chowdhury, N. Soyaib, and S. Mia, “Comparison of sliding frictions of different materials using a digital sliding friction tester,” Proceedings of International Conference on Mechanical, Industrial and Energy Engineering 2014]. For millimeter-scale glass beads moving in a fluid next to a “rough bed” of fixed smaller beads, a typical friction coefficient of 0.38 was reported [F. Charru et al., “Motion of a particle near a rough wall in a viscous shear flow,” J. Fluid Mech. 570, p. 431-453]. Therefore, we consider our value of 0.25 to be quite reasonable. We have added the above two references in the paragraph following Eq. S.3 of Supplementary Note 1.

2. The separation distance (δ) between surfaces of a particle and an optical nanofiber is the important parameter, which affects the friction coefficient. However, there is no discussion on this parameter. I think the separation distance depends on the temperature.

Answer: Particles in the evanescent field near an optical nanofiber are pulled to the nanofiber's surface by the radial optical force, thus it is reasonable to assume that a particle should end up in contact with the surface, the contact being intermittent due to the Brownian motion. This assumption has been used in our previous study on nanofiber-mediated optical manipulation of 3- μm polystyrene microspheres, ref. 42 in the manuscript. In that study, we took $\delta = 10$ nm as the mean separation due to the surface roughness (following ref. 50 where 5- μm silica and polystyrene spheres were manipulated by the evanescent field near a prism). In ref. 42, we saw an excellent agreement between the experimental data and the simulations without any adjustable parameters, hence the 10-nm separation seems to be a reasonable value. Admittedly, in that study particles were orbiting the nanofiber nearly without propulsion (although at elliptical polarizations the particles did travel along the fiber within a few-radii distance range), which might

add some hydrodynamic lift in the experiments reported in this manuscript. However, our simulations for the propulsion speed of silica particles agree with the experiments quite well, at least for nanofibers thicker than $2R_f = 700$ nm.

Compared to uncoated particles, Janus particles are different in two aspects; (i) they cannot freely rotate (but assume a certain balance orientation angle, α_0) and (ii) they can heat up significantly due to the absorption by gold. As has been shown theoretically in ref. 47, rotation may reduce the hydrodynamic lift on a particle by almost an order of magnitude. However, to the best of our knowledge, no studies have shown that uniform particles such as silica or plastic beads are indeed rotating while being optically propelled near a surface (ref. 50 assumes no rotation, by the way). As we mention in the paragraph preceding Eq.1 of the manuscript, we do not aim at a rigorous simulation of the hydrodynamic problem, but focus on optical forces and torques while the impact of friction is estimated through fitting the simplified model to the experimental data. Nevertheless, we checked how the simulation results would change if δ increases by an order of magnitude. As shown in Supplementary Figure 3 (where we added new insets to clarify the variation of α_0), the changes to optical forces and the balance angles are smaller than the confidence ranges of our measurements. Note that the increase of δ would lead to a reduction of drag and therefore an increase of the simulated propulsion speed, see Eq.3 of the main text. Since the speed is already overestimated by a factor of ~ 3 (Fig.5c-e), an increase of δ does not seem likely.

As for the thermal effects, they are certainly expected to produce both random and directional thermal forces, which might affect the effective separation distance δ . Thermophoresis of Janus particles near an optical nanofiber is a topic for a different study (as we firmly believe that experiments with tracer particles would be needed for validation of the model). Still, we have gone as far as to simulate the heat propagation in the system, see below.

3. The authors can estimate the temperature distribution under the present experimental conditions by the COMSOL software. I think that the temperature distribution might be different between silica particles and Janus particles and be dependent of the size and gold-layer-thickness. It is necessary to discuss on the temperature dependence.

Answer: We thank the reviewer for bringing up this important issue. As requested, we have simulated the temperature distribution around a silica or a Janus particle in the evanescent field near an optical nanofiber, see the new Supplementary Note 2. The temperature increase, ΔT , predicted for uncoated particles was reasonable (about 10 mK for 1 mW of optical power), but the maximum temperature calculated for gold-coated particles was surprisingly high, surpassing the boiling point of water at the optical power of about 10 mW. Being directly proportional to the power, the simulated ΔT goes beyond 500 K at 100 mW for both 10- and 20-nm thick gold coating. However, no boiling or cavitation has been recorded in our experiments with free-to-move Janus particles. They would escape from the optical tweezers when the power exceeded 2-3 mW and – once trapped by the nanofiber – would quickly reorient so that the gold cap was facing away from the fiber surface. Conversely, a stuck particle would cause cavitation if the gold cap is placed in the optical tweezers beam with the optical power of 10-20 mW.

In order to understand such a major overestimation of ΔT for Janus particles, we recall that our numerical model treats the coating as a perfectly smooth layer of crystalline gold shaped into a hemispherical cap with uniform thickness and the base radius equals the radius of the particle; $R_{\text{cap}} = R_p$. In practice, this coating is formed by vapor deposition where the gold atoms land on

the curved surface which is neither perfectly smooth, nor chemically treated for an optimum adhesion (see the new ref. 10 in the Supplementary). As a result, we expect the coating to be exceedingly rough toward the edges of the cap. In fact, our SEM images (Fig.1a, b) show some surface and edge irregularities. They may significantly alter the coating properties, in particular the efficiency of surface plasmon propagation, absorption of light and the associated heating. Our 3D model is too coarse for adequate simulations of such small features, and we cannot further refine the mesh because of the computational limits. The only thing we could do in order to mimic the deterioration of the coating toward its edges was to reduce the gold cap base radius, $R_{\text{cap}} \leq R_p$, while keeping the orientation angle and material properties of the remaining gold intact. As can be seen in the new Fig.7, this reduction has a major effect on the temperature increase, as well as on the optical forces. By comparing the propulsion enhancement simulated for Janus particles with a reduced cap and the experimental results for 700-nm nanofibers (shaded bands in Fig.7g), we found a good match for the case when the cap base radius is just 6.6% smaller than the particle radius. Moreover, simulations with this R_{cap} value agree quite well with the measurements for various nanofiber thickness (see the inset in Fig.7g). Importantly, no boiling is expected for such a reduced cap.

Although this model produces more realistic results compared with that described in Fig.2, we still consider it as a “toy” model, because it neglects the polycrystallinity and surface roughness of the coating. The important added value of these simulations is that the geometry commonly used for simulations of Janus particles is not accurate, especially when one considers the heat propagation.

Another note. At the end of the original manuscript, we mentioned the propulsion test performed at an optical power of 200 mW, assuming that the propulsion enhancement would stay constant as it does in the 20-100 mW range. No cavitation was observed at 200 mW, although the thermal model predicts the temperature to exceed the boiling point of water, even for the gold cap radius reduced by 6.6%. The test at 200 mW was performed more than 1.5 years ago and we could not find video recordings and thus verify the propulsion speed. In addition, at that time we did not check if the particle was moving along the thinnest part of the fiber, and the subsequent measurements of the fiber with SEM has not been implemented yet. Thus, the particle could be near a thicker section of fiber where the field intensities, propulsion speeds, and heat generation were lower than expected. In any case, the statement about the 200 mW test cannot be verified, hence we removed it from the revised manuscript. The currently claimed top propulsion speed of “71 body length per second for the transmitted power of 100 mW” is verifiable as we have the source video recordings and the SEM measurements of the nanofiber size.

4. The thermal fluctuation is also important for analysis of the particle motion in liquid. The temporal variation of the particle position affects the separation distance (δ) and the friction coefficient (μ) and then the orientation angle (α). I think the author should discuss on this issue.

We agree that Brownian motion may have a significant effect on the optomechanics of Janus particles near a nanofiber. However, in order to simulate this random motion, as well as the thermophoresis, one must know the actual temperature distribution in the system (and the relevant temperature-dependent parameters such as viscosity, Debye length, zeta potential). As we have shown, the commonly used geometry with a smooth hemispherical gold cap leads to completely unrealistic temperature values. The reduced cap model we tried is also not very accurate, and it shows major changes in the temperature distribution (compare the position of the hottest point in Fig.7a and Fig.7b). Such changes would affect the local thermal forces which are

known to depend on the temperature gradient along the particle surface. An accurate thermal picture could be obtained only by experiments with tracer particles, which are certainly beyond the scope of our study.

Following the reviewer's comment, we have explored the role played by orientation angle of the gold cap in the temperature distribution. We found that an increase in the gold cap tilt by 1 degree corresponds to an increase in ΔT by approximately 5-7% (see the new Supplementary Figure 4f, g). In fact, all the considered parameters – balance orientation angle, gold cap base radius, separation distance, friction coefficient – are linked in the model and potentially could be adjusted independently to find the best match between the simulations and the experimental data. However, 3D numerical simulations even with these rather coarse meshes are very time consuming and such a multiparameter optimization is not reasonable at the current level of computational resources. Therefore, we chose a single adjustable parameter – the friction coefficient – in the main optical model in order to match the predicted balance orientation angles with the experimentally observable ones.

Reviewer 2

The revised version has significantly improved, in the response to the previous reports the authors have done additional experiments and clarified various issues. The authors have made quite an effort to confirm their model for the motion along the fiber. Now this is a sound and rather complete work on optical manipulation of JP.

The experimental setup is rather well known from previous studies of colloidal particles, original results concern characterizing and understanding particle motility. The overall picture is rather convincing, even if important discrepancies persist regarding the torques and directional enhancement factors.

Though this is a nice paper, I am not entirely convinced it warrants publication in Nature Communications.

Answer: We thank the reviewer for the positive feedback. True, optical manipulation with nanofibers and finite-element based simulations of light-induced effects in microscale objects are both known to the scientific community. However, the use of waveguides and particularly optical nanofibers for moving Janus particles has not been attempted prior to this work. The results are quite interesting and potentially impactful to the actively researched domain of composite metallo-dielectric particles and their applications. Therefore, we firmly believe that Nature Communications is the proper choice of the journal for publication. Hopefully, the newly added thermal model (in Supplementary) and Fig.7 (main text) help to further justify this choice.

REVIEWERS' COMMENTS

Reviewer #1 (Remarks to the Author):

In the revised manuscript, the authors discussed the temperature distribution and thermal effects under their experimental conditions, according to the reviewer's comments. I think that their discussions are not enough clear for elucidation of the observed results. However, I understand that those analyses are beyond the scope of this paper. They have successfully discussed on optical forces and torques exerted on silica microspheres half-coated with nanometer-thick layer of gold. Thus, I would recommend that the manuscript can be accepted for publication.

Manuscript ID: NCOMMS-21-32539B

“Evanescent field trapping and propulsion of Janus particles along optical nanofibers”

by Georgiy Tkachenko, Viet Giang Truong, Cindy Liza Esparlas, Isha Sanskriti, and Síle Nic Chormaic.

Response to the reviewers' reports on the revised version

Reviewer 1

In the revised manuscript, the authors discussed the temperature distribution and thermal effects under their experimental conditions, according to the reviewer's comments. I think that their discussions are not enough clear for elucidation of the observed results. However, I understand that those analyses are beyond the scope of this paper. They have successfully discussed on optical forces and torques exerted on silica microspheres half-coated with nanometer-thick layer of gold. Thus, I would recommend that the manuscript can be accepted for publication.

Answer: We thank the reviewer for the careful consideration of our work. We agree that the thermal model is far from being perfect. Still, we believe that the experimental evidence, the optomechanical simulations, and the hints obtained from the thermal simulations will be interesting and useful for the readers. We are glad that the reviewer supports the publication of our manuscript.